# Sarcopenia and Mortality in Older Hemodialysis Patients

**DOI:** 10.3390/nu14112354

**Published:** 2022-06-05

**Authors:** M. Luz Sánchez-Tocino, Blanca Miranda-Serrano, Antonio López-González, Silvia Villoria-González, Mónica Pereira-García, Carolina Gracia-Iguacel, Isabel González-Ibarguren, Alberto Ortíz-Arduan, Sebastian Mas-Fontao, Emilio González-Parra

**Affiliations:** 1Fundación Renal Íñigo Álvarez de Toledo, 28003 Madrid, Spain; lsanchez@friat.es (M.L.S.-T.); blanca.miranda.serrano@gmail.com (B.M.-S.); svilloriagonzalez@friat.es (S.V.-G.); mpereira@friat.es (M.P.-G.); 2Servicio de Hemodialisis, Complejo Hospitalario Universitario A Coruña, 15006 A Coruna, Spain; antonio.lopez.gonzalez@sergas.es; 3Servicio de Nefrología e Hipertensión, Fundación Jiménez Díaz, 28040 Madrid, Spain; cgraciai@quironsalud.es (C.G.-I.); aortiz@fjd.es (A.O.-A.); 4Servicio de Geriatria, Hospital Universitario de Guadalajara, 19002 Guadalajara, Spain; isagibar@gmail.com; 5CIBERDEM, IIS-Fundación Jiménez Díaz, 28040 Madrid, Spain

**Keywords:** sarcopenia, EWGSOP2, elderly, hemodialysis, kidney replacement therapy, mortality, kidney failure

## Abstract

(1) Sarcopenia is a progressive loss of skeletal muscle mass and strength. The aim of this study was to determine the association of sarcopenia, defined according to the Working Group on Sarcopenia in Older People (EWGSOP2) diagnostic criteria, with mortality at 24 months in very elderly hemodialysis patients. (2) A prospective study was conducted in 60 patients on chronic hemodialysis who were older than 75 years. Sarcopenia was diagnosed according to EWGSOP2 criteria. Additionally, clinical, anthropometric and analytical variables and body composition by bioimpedance were assessed. The date and cause of death were recorded during 2 years of follow-up. (3) Among study participants, 41 (68%) were men, the mean age 81.85 ± 5.58 years and the dialysis vintage was 49.88 ± 40.29 months. The prevalence of probable sarcopenia was 75% to 97%, depending on the criteria employed: confirmed sarcopenia ranged from 37 to 40%, and severe sarcopenia ranged from 18 to 37%. A total of 30 (50%) patients died over 24 months. Sarcopenia probability variables were not related to mortality. In contrast, sarcopenia confirmation (appendicular skeletal muscle mass, ASM) and severity (gait speed, GS) variables were associated with mortality. In multivariate analysis, the hazard ratio (95% confidence interval) for all-cause death was 3.03 (1.14–8.08, *p* = 0.028) for patients fulfilling ASM sarcopenia criteria and 3.29 (1.04–10.39, *p* = 0.042) for patients fulfilling GS sarcopenia criteria. (4) The diagnosis of sarcopenia by EWGSOP2 criteria is associated with an increased risk of all-cause death in elderly dialysis patients. Specifically, ASM and GS criteria could be used as mortality risk markers in elderly hemodialysis patients. Future studies should address whether the early diagnosis and treatment of sarcopenia improve outcomes.

## 1. Introduction

Sarcopenia is a progressive loss of skeletal muscle mass and strength that occurs primarily with advancing age [1]. The process starts from age 40 years, but it accelerates after age 70, and more than half of those aged over 80 years have sarcopenia [2]. However, the prevalence of sarcopenia is variable and depends on the diagnostic criteria used. A consensus definition of sarcopenia was established in 2010 by the European Working Group on Sarcopenia in Older People (EWGSOP) [3]. This definition was recently revised (EWGSOP2), incorporating novel evidence from the last 10 years [4].

In hemodialysis (HD), the prevalence of sarcopenia ranges from 4% to 64% [5,6,7]. In our very elderly population on haemodialysis, the prevalence of confirmed sarcopenia according to EWGSOP2 is 40% [8]. In chronic kidney disease (CKD), protein catabolism induced by metabolic acidosis, accumulation of uraemic toxins and pro-inflammatory cytokines, and the dialysis procedure itself can lead to an accelerated degradation of lean mass [9]. In addition, increasing age, decreased intake, increased nutrient losses in dialysate and comorbidities leading to inactivity and hospitalisation increase the incidence of sarcopenia [10].

Sarcopenia was defined as an independent condition in 1994 [11]. It is a key cause of frailty and dependence in older subjects [12,13] and is associated with poor endurance, physical inactivity, slow gait and decreased mobility [14,15]; an increased risk of disability and higher health care costs [16] and mortality [17,18,19].

While initially the concept of sarcopenia focused on muscle size, muscle strength does not depend on muscle size alone, and the two entities can be separate [20,21]. Indeed, the loss of strength is greater than the loss of muscle mass in older patients, and strength can decrease while muscle size is still maintained or even increased [22].

The recommended criteria for defining sarcopenia in the healthy elderly population have not been sufficiently studied in elderly HD patients. Specifically, whether low strength and/or muscle wasting diagnosed according to EWGSOP2 is associated with mortality in this specific population has not been previously explored. Thus, the association of muscle strength or muscle wasting with mortality should be analysed independently, as they may have different clinical implications [23].

The aim of this study was to assess the association of sarcopenia diagnosed according to EWGSOP2 diagnostic (loss of strength, muscle mass and function) with 24-month mortality in very elderly haemodialysis patients.

## 2. Materials and Methods

A prospective study was conducted in patients on chronic haemodialysis in three outpatient centres and one hospital unit of the Fundación Renal Íñigo Álvarez de Toledo in Spain. It was approved by the ethics committee of the Hospital Universitario Fundación Jiménez Díaz (act n° 03/19) and complied with the standards recognized by the Declaration of Helsinki of the World Medical Association, as well as the Standards of Good Clinical Practice, in addition to compliance with Spanish legislation on biomedical research (Law 14/2007). All participants signed an informed consent for their participation.

The study ran from February 2019 to February 2021. Physical tests for the diagnosis of sarcopenia were performed in February 2019. The inclusion criteria were age between 75 and 95 years, able to perform physical fitness assessment tests or dynamometry, had been on HD for more than 3 months and had signed the consent form.

### 2.1. Primary Variables

The primary variables were those needed to assess sarcopenia diagnostic according to EWGSOP2 [4], comprising four stages. Cut-off values are listed in Table 1. The stages are:**Clinical suspicion/case finding:** Defined by the SARC-F Survey Score. SARC-F (Strength, Assistance walking, Rise from a chair, Climb stairs, and Falls) [24].**Probability: loss of strength.** The probability of sarcopenia can be defined by either of the following two variables: (1) upper limb: handgrip strength assessed by dynamometry (GSD) using an electric dynamometer (CAMRY^®^ Model EH101) [25]; (2) lower limb: test sit to stand to sit 5 (STS-5) [26].**Confirmation: muscle mass**. Defined by appendicular skeletal muscle mass (ASM), measured by bioimpedance (BIA). A MALTRON^®^ brand BioScantouch i8 BIA was used. The assessments were performed in the second session of the week between the first and second hour of HD, given that the device allows intrahaemodialytic measurements.**Severity: physical condition.** The severity of sarcopenia was defined by any of the following three variables: (1) gait speed (GS), measured as time needed to walk 4 m expressed in meters per second [27]; (2) the Timed-Up and Go test (TUG) [28]; (3) the Short Physical Performance Battery (SPPB) [29].

**Table 1 nutrients-14-02354-t001:** Cut-off points for diagnosis of sarcopenia according to EWGSOP2.

Diagnostic Stages	Test	Males	Female
Finding	SARC-F [24]	≥4 points	≥4 points
Evaluation	Upper limbs: GSD [30]	<27 kg	<16 kg
	Lower limbs: STS5 [31]	>15 s	>15 s
Confirmation	AMS [32]	<20 kg	<15 kg
Severity	GS [32,33]	≤0.8 m/s	≤0.8 m/s
	TUG [34]	≥20 s	≥20 s
	SPPB [35,36]	≤8 points	≤8 points

SARC-F: sarcopenia screening survey, GSD: grip strength by dynamometry, STS-5: sit-to-stand-to-sit 5test, AMS: muscle mass, GS: gait speed, TUG: the Timed-Up and Go test, SPPB: the Short Physical Performance Battery.

Strength measurements and physical tests were performed on the second day of the week before the dialysis session. This avoided post-dialysis fatigue.

### 2.2. Other Variables

Additionally, clinical variables (sex, age, cause of renal disease, dialysis vintage), anthropometric variables (height, weight and body mass index (BMI); arm, waist and hip circumference; waist-hip index (WHI); and tricipital, abdominal and subscapular folds), analytical variables (serum albumin, total proteins, haemoglobin, haematocrit, C reactive protein (CRP), 25-OH vitamin D), dialysis efficacy (Daurgidas Kt/Vurea) and body composition by bioimpedance were assessed. The date and cause of death were recorded. Observer, anthropometric and physical tests were performed by the same physical activity and sport sciences professionals.

### 2.3. Statistics

Statistical analysis was performed with IBM SPSS Statistics V20 and R. Quantitative variables were presented as mean and standard deviation. Qualitative variables were presented as absolute numbers and percentages. Student’s *t* test was used to compare the quantitative variables. The associations between the qualitative variables were assessed using the chi-square test. The level of statistical significance was *p* less than or equal to 0.05.

Survival analysis was performed with the Kaplan-Meier survival curve and the log-rank test. In addition, the Cox proportional hazard model was used to determine hazard ratios with a 95% confidence interval. For mortality analysis, a single test was used in each EWGSOP2 stage. For probable sarcopenia, GSD was chosen, and for severe sarcopenia, GS was used, based on their simplicity and reproducibility and the combined representation of upper and lower limbs [30]. In a sensitivity analysis, COVID-19 deaths were censored.

## 3. Results

### 3.1. Participants

A total of 60 patients participated in the study, and 41 (68%) were men, with mean age 81.85 ± 5.58 years and dialysis vintage 49.88 ± 40.29 months. The causes of renal disease were diabetes mellitus (28%), unknown cause (32%), hypertension (20%), interstitial nephritis (7%), glomerulonephritis (5%) and others (8%). Table 2 shows baseline characteristics.

### 3.2. Prevalence of Sarcopenia According to EWGSOP2

Table 3 shows the analysis of the sarcopenia variables, the frequency of each individual diagnostic criterion and the prevalence of the diagnostic criteria established by the combination of variables according to the EWGSOP2 algorithm. A low muscle mass measured by BIA used to confirm sarcopenia is less prevalent than the severity criteria (40% vs. 70–75%).

The prevalence of probable sarcopenia was 75% as assessed by GSD, 88% by STS-5 and 97% when fulfilling either the GSD or the STS-5 criteria. The prevalence of confirmed sarcopenia ranged from 37 to 40% depending on the probability criteria used. The prevalence of severe sarcopenia ranged from 18 to 37%.

### 3.3. Mortality

A total of 30 (50%) patients died during 24 months of follow-up: 22 (54%) men and 8 (42%) women (*p* = 0.405). The causes of death were 13/30 (43%) cardiovascular, 10/30 (33%) infection, 3/30 (10%) digestive, 2/30 (7%) malignancy and 2/30 (7%) trauma. COVID-19 deaths represented 5/10 (50%) of infection deaths and 5/30 (16.5%) of total deaths.

Table 4 summarizes the data for the surviving and deceased patients (while in Appendix A the population is broken down by sex). Patients who died were older, had higher incidence of cardiovascular disease and had higher CRP levels than those that survived.

Mortality according to the individual EWGSOP2 sarcopenia criteria chosen for analysis is shown in Table 5. The Find (SARC-F) and probability variables (GSD and STS5) were not related to mortality. In contrast, the confirmatory variables (ASM) and the variables marking severity (GS, TUG, SPPB) were associated with mortality.

Figure 1 plots the survival curves for all-cause mortality according to the EGWSOP2 criteria over two years of follow-up. Probable sarcopenia (GSD) cases were not associated with increased mortality (Figure 1A), whereas confirmation criteria (GSD+ AMS) (Figure 1B) and severity criteria (GSD, AMS and GS) (Figure 1C) were associated with increased all-cause mortality.

The hazard ratio (95% confidence interval) for all-cause death at 24 months was 2.5 (1.13–5.55, *p* = 0.024) for patients fulfilling AMS criteria for confirmed sarcopenia and 3.02 (1.09–8.32, *p* = 0.033) for patients fulfilling GS criteria for severe sarcopenia (Figure 2A). Several sarcopenia diagnostic criteria were explored as univariate cox regression in Appendix A.

Likewise, when all-cause mortality is analysed by Cox regression, including together the two main factors that can affect sarcopenia, age and the presence of cardiovascular disease, in the analysis, all factors are statistically significant in a Cox univariate analysis: ASM criteria HR 2.86 (1.29–5.57, *p* = 0.008); gait speed criteria HR 3.24 (1.24–8.49, *p* = 0.0017); cardiovascular disease HR 3.55 (1.36–9.30, *p* = 0.01) and age HR 1.11 (1.04–1.18, *p* = 0.002). When these data are taken together in a multivariate analysis (Figure 2): The ASM and GS criteria retain their statistical significance when the three EGWSOP2 criteria are considered (Figure 2A), and loss of muscle mass, strength and physical performance are related to mortality, although, age and CVD attenuated this association in multivariable analysis. (Figure 2B).

In a sensitivity analysis, COVID-19 deaths were censored (Appendix A). Probable sarcopenia (GSD) cases were not associated with increased non-COVID-19 mortality (Appendix A), but the confirmation criteria (GSD + AMS) (Appendix A) and severity criteria (GSD, AMS and GS) (Appendix A) were associated with increased non-COVID-19 mortality. The hazard ratio (95% confidence interval) for non-COVID-19 death at 24 months was 3.00 (1.19–7.33, *p* = 0.020) for patients fulfilling AMS criteria for confirmed sarcopenia and 3.62 (1.17–11.24, *p* = 0.026) for patients fulfilling GS criteria for severe sarcopenia (Appendix A).

## 4. Discussion

The main finding of this study was that a diagnosis of sarcopenia by EWGSOP2 criteria significantly increases the risk of death in elderly haemodialysis patients. The novelty lies in the assessment of the performance of the different components of the recently updated EWGSOP2 diagnostic algorithm for sarcopenia in a specific population group that has been little studied: very elderly patients on haemodialysis [31]. Specifically, the predictive value of the algorithm for all-cause mortality was explored. Loss of strength has been related to mortality in haemodialysis patients, and loss of muscle mass has also been studied [5,32], but the EWGSOP2 diagnostic algorithm combines the loss of both muscle mass and strength, and its performance should be validated in different patient populations.

According to the EWGSOP2 criteria in our population over 75 years old on haemodialysis, the diagnosis of sarcopenia by measuring muscle mass using bioimpedance was confirmed in 38% [8]. Using the EWGSOP2 criteria, other authors have found a bioimpedance diagnosis of 60% in elderly populations without renal disease [33]. It is challenging to understand the differences as renal patients have multiple factors that promote the loss of muscle mass and strength. This may be due in part to changes in body composition, which decrease the accuracy of BIA due to water overload [34]. However, the reproducibility of the BIA has been proven in previous studies [8], and ASM estimates made by tetrapolar BIAs are consistent with other studies. In our study, the decrease in muscle mass measured by BIA was milder than the loss of muscle strength and function. Thus, 75% of patients had a loss of strength measured by dynamometry, of whom only 38% were confirmed to have sarcopenia by BIA, and almost all of them (32% of the total) had severe sarcopenia. The discrepancy between mass and strength in CKD patients is consistent with prior reports [5].

BIA is an easily accessible and inexpensive method for measuring muscle mass in patients on renal replacement therapy [35]. Thus, we previously showed that the decrease in muscle mass assessed by BIA is a good marker of mortality in patients with CKD [32,36]. However, in CKD patients loss of muscle mass better predicted outcomes when assessed dynamically, i.e., change over time. Our current study shows that the exclusive use of BIA to diagnose sarcopenia misses patients with loss of muscle strength. Overhydration may alter the estimation of muscle mass by BIA in haemodialysis patients and therefore reduce the number of patients classified as sarcopenic [8,9]. However, our results also show that a sarcopenia diagnosis using only ASM as assessed by BIA and the cut-off points set by EWGSOP2 is a good predictor of mortality, likely not only due to objective loss of muscle mass but also due to overhydration [37] or body composition [38]. Thus, despite ASM not being a sensitive marker of sarcopenia in elderly dialysis patients, the EWSGOP2 cut-off values for ASM do predict all-cause mortality.

We have found an overall 50% two-year mortality rate in haemodialysis patients over 75 years of age. The overall mortality annual rate in our HD unit is 9.2% [39]. Mortality in very elderly patients is higher in different publications, reaching 35% per year [40].

Sarcopenia has been linked to mortality, disability, frailty and admissions in patients with kidney disease [41,42,43,44]. Sarcopenia defined as reduced handgrip strength and low skeletal muscle mass index estimated by BIA was an independent predictor of mortality in these patients [18,36]. However, the EWGSOP2 criteria had not been previously assessed for prediction of mortality in very elderly patients on haemodialysis. The risk of mortality associated to sarcopenia in elderly patients on dialysis is highly influenced by age and cardiovascular disease. Thus, adjustment for these two variables resulted in a persistent numerical increase in risk of all-cause death, hovering around two-fold, that was no longer statistically significant in our analysis.

Although loss of grip strength has been related to increased mortality in different studies, we did not observe an association with mortality when applying the EWGSOP2 criteria to a very elderly haemodialysis population. However, the EWGSOP2 criteria for muscle mass and gait speed were associated with an increased risk of all-cause death, i.e., loss of muscle mass or function were associated with an increased risk of death. Although this was an observational study and did not assess the impacts of interventions in patients with sarcopenia on the risk of death, the high risk of death within a relatively short follow-up period suggests that by the time that sarcopenia is confirmed by the loss of muscle mass as assessed by bioimpedance, the margin for intervention that decreases the risk of death is minimal. Thus, we suggest that efforts be directed at earlier detection and intervention to prevent, rather than to treat, the loss of muscle mass, likely when loss of strength is detected. This hypothesis should be evaluated in prospective interventional studies.

In a population with normal renal function, severe sarcopenia diagnosed using EWGSOP2 was associated with an increased risk of death compared with people without sarcopenia when using EWGSOP2 (HR 4.11, 95% CI 1.88–9.00). Consistent with our findings, older adults with decreased gait speed had a 76% higher risk of dying (*p* = 0.033) [45]. Mortality is also increased in otherwise healthy elderly persons with decreased grip strength, classified as probable sarcopenia [46]. In patients with kidney failure, an increased mortality in patients with decreased grip strength has also been reported in a younger population [47] and in haemodialysis [36]. In our very elderly haemodialysis population, we did not find an association of decreased grip strength with increased risk of death, likely to the high prevalence of low muscle strength (75% and 88% for upper and lower limb strength, respectively). We hypothesize that intervention should be started at this earlier stage of decreased muscle strength, before the loss of muscle mass is detected by bioimpedance is observed [31,48]. In this regard, sarcopenia is a reversible disorder and early diagnosis and treatment is one of the most effective tools for recovering from sarcopenia and reducing mortality [49].

Some limitations must be acknowledged. Data were not available for some relevant variables such as cachexia, peripheral vascular disease or physical activity. The number of patients was not high as many of the tests are complex to perform in such an elderly population. Furthermore, only elderly patients were studied, and these results cannot be generalise to other population groups. In future studies, it would be interesting to analyse other age groups.

## 5. Conclusions

In conclusion, the diagnosis of sarcopenia by EWGSOP2 increases the risk of all-cause mortality in elderly haemodialysis patients. Specifically, ASM and GS criteria could be used as mortality risk markers in elderly haemodialysis patients. Future studies should address whether early diagnosis and treatment of sarcopenia improves outcomes.

## Figures and Tables

**Figure 1 nutrients-14-02354-f001:**
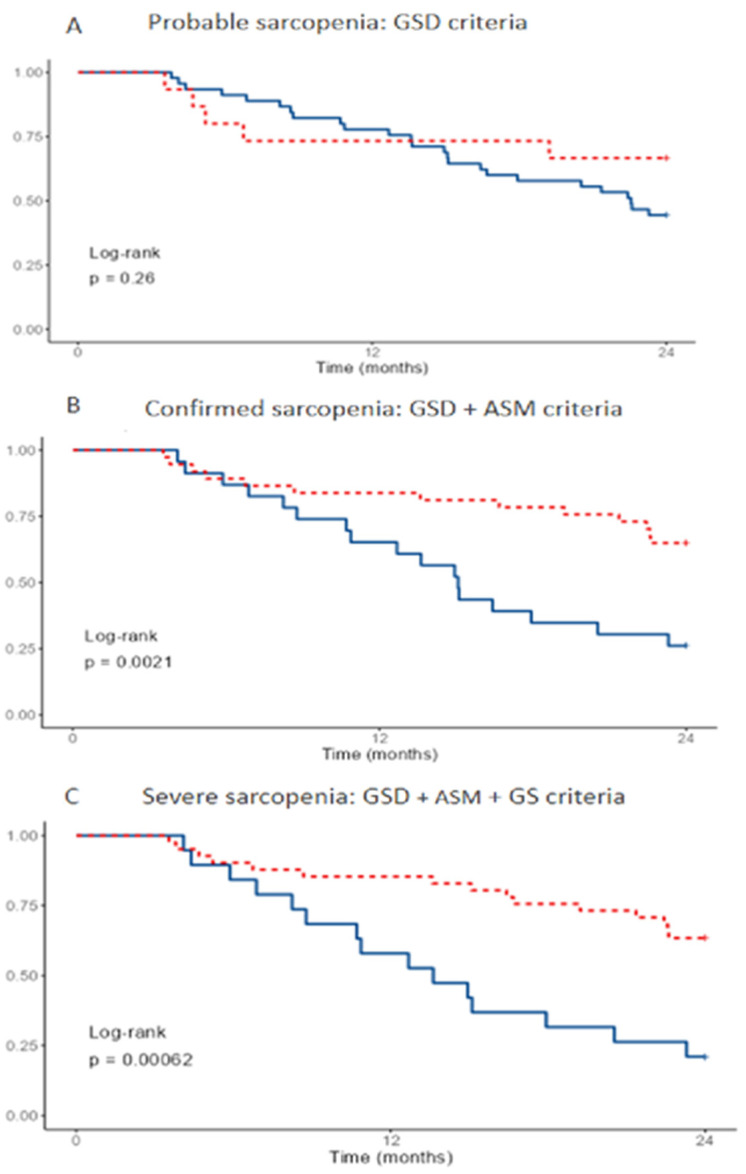
Survival according to the EGWSOP2 criteria at 24 months. Vertical axis shows survival. Probable of severe sarcopenia in blue, no sarcopenia in discontinuous red. Survivors, n = 30. Deceased, n = 30. GSD: grip strength by dynamometry, ASM: appendicular skeletal muscle mass, GS: gait speed.

**Figure 2 nutrients-14-02354-f002:**
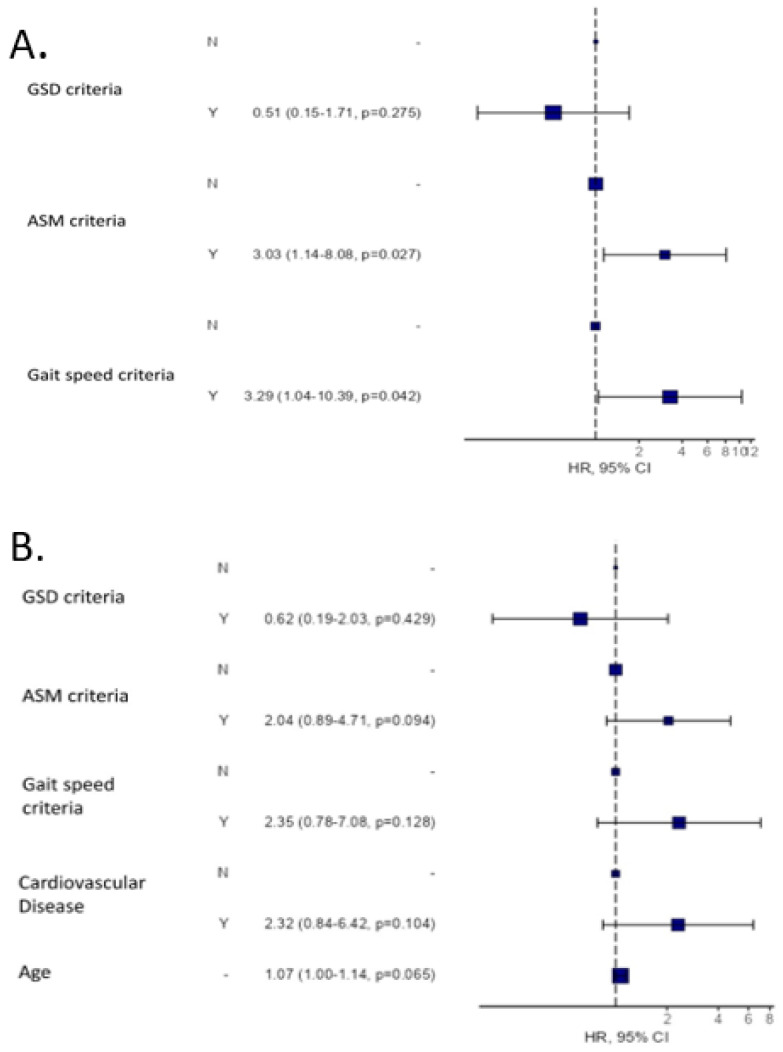
All-cause mortality risk analysis according to the EGWSOP2 criteria at 24 months. (**A**) Univariate analysis. (**B**) Multivariate analysis including the EGWSOP2 criteria, age and cardiovascular disease.

**Table 2 nutrients-14-02354-t002:** Demographic, analytical, anthropometric and body composition (bioimpedance) data expressed as mean ± SD, median (interquartile range) or n (%).

	All (n = 60)	Male (n = 41)	Female (n = 19)	* *p* Value
*Demographic data*				
Age (years)	81.85 ± 5.58	81.31 ± 5.72	83 ± 5.22	0.27
Dialysis vintage (months)	36 (21–67)	42 (21–62)	36 (21–80)	0.78
*Comorbidities*				
Diabetes	24/60 (40%)	18/41 (44%)	6/19 (32%)	0.365
Cardiovascular disease	40/60 (67%)	32/41 (78%)	8/19 (42%)	**0.006**
Malignancy	21/60 (35%)	17/41 (41%)	4/19 (21%)	0.123
*Analytical data*				
Albumin (g/dL)	3.66 ± 0.47	3.69 ± 0.41	3.59 ± 0.60	0.43
Hemoglobin (g/dL)	11.26 ± 1.13	11.40 ± 1.02	11.26 ± 1.13	0.63
C Reactive Protein (mg/L)	0.74 (0.33–1.62)	0.82 (0.39–1.65)	0.74 (0.26–1.50)	0.82
25OH Vitamin D_3_ (ng/mL)	21.51 ± 13.13	21.14 ± 12.95	22.31 ± 13.81	0.75
Kt/V_urea_	1.80 ± 0.38	1.70 ± 0.372	2.01 ± 0.30	**0.002**
*Anthropometric data*				
Body mass index (kg/m^2^)	25.20 ± 3.64	25.73 ± 3.40	24.08 ± 3.95	0.10
Mid-Upper Arm Circumference (cm)	25.69 ± 3.11	26.62 ± 2.66	25.69 ± 3.11	0.26
Waist Perimeter (cm)	92.78 ± 10.41	97 ± 7.58	83.87 ± 10.07	**<0.001**
Hip Perimeter (cm)	100.57 ± 7.23	101.61 ± 6.92	98.39 ± 7.56	0.121
Waist hip index	0.92 ± 0.08	0.95 ± 0.69	0.85 ± 0.06	**<** **0.001**
Tricipital Fold (mm)	11.89 ± 4.32	10.86 ± 3.83	14.08 ± 4.60	**0.008**
Abdominal Fold (mm)	18.38 ± 6.35	19.79 ± 6.98	15.76 ± 4,39	0.054
Subscapular Fold (mm)	15.39 ± 7.09	16.42 ± 6.69	13.22 ± 7.61	0.115
*Body composition*				
Muscle Mass (kg)	19.27 ± 3.82	20.98 ± 3.22	15.57 ± 1.87	**<0.001**
Fast Mass (kg)	22.91 ± 5.07	22.91 ± 5.07	22 ± 7.40	0.581
Total Body Water (L)	32.41 ± 6.52	35.52 ± 5.17	25.69 ± 3.15	**<0.001**
Overhydration (L)	1. 10 ± 1.41	1.26 ± 1.61	0.75 ± 1.21	0.233

* *p* < 0.05 in bold.

**Table 3 nutrients-14-02354-t003:** The mean values of the sarcopenia marker variables, the frequency of sarcopenic individuals for each variable and the prevalence of sarcopenia according to the EWGSOP2 diagnostic algorithm. n = 60. Data presented as mean ± SD or n (%).

Individual Criterion	Value for Criterion	Consistent with Sarcopenia, n (%)	Diagnostic Algorithm	Consistent with Sarcopenia, n (%)
*Find*				
SARC-F (points)	2.6 ± 2.3	18 (30%)		
Assess: Probable sarcopenia				
GSD (Kg)	19.2 ± 6.6	45 (75%)	GSD	45 (75%)
STS5 (s)	20.3 ± 6.3	53 (88%)	STS-5	53 (88%)
			GSD and/or STS-5	58 (97%)
Confirm: Sarcopenia confirmed				
ASM (kg)	19.3 ± 3.8	24 (40%)	GSD + ASM	23 (38%)
			STS-5 + ASM	22 (37%)
			GSD and/or STS-5 + ASM	24 (40%)
*Severity:* Severe sarcopenia				
GS (m/s)	0.69 ± 0.27	42 (70%)	GSD + ASM + GS	19 (32%)
			STS-5 + ASM + GS	19 (32%)
			GSD and/or STS-5 + ASM + GS	19 (32%)
TUG (s)	19.1 ± 12.1	22 (37%)	GSD + ASM + TUG	11 (18%)
			STS-5 + ASM + TUG	11 (18%)
			GSD and/or STS5 +ASM + TUG	11 (18%)
SPPB (points)	6.2 ± 2.9	45 (75%)	GSD + ASM + SPPB	21 (35%)
			STS-5 + ASM + SPPB	21 (35%)
			GSD and/or STS-5 + ASM + SPPB	21 (35%)
			GSD and/or STS-5 + ASM + GS and/or TUG and/or SPPB	22 (37%)

SARC-F: Strength, Assistance walking, Rise from a chair, Climb stairs, and Falls; GSD: grip strength by dynamometry, STS-5: sit to stand to sit 5, ASM: appendicular skeletal muscle mass, GS: gait speed, TUG: Timed-Up and Go test, SPPB: Short Physical Performance Battery. Variables used in the mortality analysis are marked in orange.

**Table 4 nutrients-14-02354-t004:** Characteristics of patients alive or deceased at 2 years. Data shown as mean ± SD, n (%).

	Alive, n = 30	Deceased, n = 30	*p* Value
*Demographic data*			
Age (years)	79.9 ± 4.9	83.8 ± 5.6	**0.005**
Dialysis vintage (months)	47.9 ± 42.3	51.6 ± 39.1	0.728
*Comorbidities*			
Diabetes	11.0 (36.7%)	13.0 (43.3%)	0.598
Cardiovascular disease	15.0 (50.0%)	25.0 (83.3%)	**0.006**
Malignancy	9.0 (30.0%)	12.0 (40.0%)	0.417
*Analytical data*			
Albumin (g/dL)	3.7 ± 0.4	3.6 ± 0.5	0.233
Hemoglobin (g/dL)	11.5 ± 1.0	11.2 ± 1.1	0.225
C Reactive Protein (mg/L)	0.7 ± 0.7	2.5 ± 3.7	**0.013**
25OH Vitamin D_3_ (ng/mL)	19.3 ± 10.2	23.8 ± 15.4	0.194
Kt/V_urea_	1.8 ± 0.4	1.8 ± 0.3	0.533
*Anthropometric data*			
Body mass index (kg/m^2^)	25.5 ± 4.3	24.9 ± 2.9	0.518
Mid-Upper Arm Circumference (cm)	26.7 ± 3.2	25.9 ± 2.4	0.338
Waist Perimeter (cm)	92.2 ± 11.2	93.4 ± 9.7	0.689
Hip Perimeter (cm)	100.1 ± 8.3	101.1 ± 6.1	0.607
Waist hip index	0.9 ± 0.1	0.9 ± 0.1	0.828
Tricipital Fold (mm)	1.2 ± 0.5	1.2 ± 0.4	0.637
Abdominal Fold (mm)	18.9 ± 7.1	17.7 ± 5.3	0.558
Subscapular Fold (mm)	15.6 ± 7.9	15.1 ± 6.2	0.788
*Body composition*			
Muscle Mass (kg)	20.0 ± 4.5	18.6 ± 2.9	0.150
Fast Mass (kg)	22.9 ± 7.1	22.3 ± 4.4	0–703
Total Body Water (L)	33.0 ± 7.4	31.8 ± 5.5	0.488
Overhydration (L)	0.9 ± 1.6	1.2 ± 1.4	0.584

Note: *p* < 0.05 in bold.

**Table 5 nutrients-14-02354-t005:** Mortality at 2 years according to each individual variable in the EWGSOP2 diagnostic algorithm. Data expressed as n (%).

	Sarcopenia	Alive, n = 30	Deceased, n = 30	*p* Value
*Find*				
SARC-F (points)	YES, n = 20	10/20 (50%)	10/20 (50%)	1.000
	NO, n = 40	20/40 (50%)	20/40 (50%)	
*Assess*				
GSD (Kg)	YES, n = 15	10/15 (66%)	5/15 (33%)	0.136
	NO, n = 45	20/45 (44%)	25/45 (56%)	
STS5 (s)	YES, n = 53	25/53 (47%)	28/53 (53%)	0.227
	NO, n = 7	5/7 (71.5%)	2/7 (28.5%)	
*Confirm*				
ASM (kg)	YES, n = 24	7/24 (29%)	17/24 (71%)	**0.008**
	NO, n = 36	23/36 (64%)	13/36 (36%)	
*Severity*				
GS (m/s)	YES, n = 42	15/42 (36%)	27/42 (64%)	**>0.001**
	NO, n = 18	15/18 (83%)	3/18 (17%)	
TUG (s)	YES, n = 22	5/22 (23%)	17/22 (77%)	**0.001**
	NO, n = 38	25/38 (66%)	13/38 (33%)	
SPPB (points)	YES, n = 45	19/45 (42%)	26/45 (58%)	**0.036**
	NO, n = 15	11/15 (73%)	4/15 (27%)	

SARC-F: Strength, Assistance walking, Rise from a chair, Climb stairs, and Falls; GSD: grip strength by dynamometry, STS-5: sit to stand to sit 5, ASM: appendicular skeletal muscle mass, GS: gait speed, TUG: Timed-Up and Go test, SPPB: Short Physical Performance Battery. Note: *p* < 0.05 in bold.

## Data Availability

The database employed for the current study is available upon request.

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
