# Peer review of "Sarcopenia and Mortality in Older Hemodialysis Patients"

_nutrients, 2022, doi:10.3390/nu14112354_

Round 1
Reviewer 1 Report
This paper entitled ” Sarcopenia according to EWGSOP2 and mortality in older hemodialysis patients” demonstrated that evaluation and definition of sarcopenia according to EWGSOP2 can predict the mortality in elderly patients with hemodialysis. This paper looks interesting, but there seems several questions remain to be resolved.
Major concern,
- Authors should highlight novelty of the present manuscript.
- Authors enrolled sixty hemodialysis patients in the present study. Average age of the patients is 81.85 years old which is higher than the men’s life expectancy (79.59 yo) in Spain. Assessment of sarcopenia in the patients seems to provide interesting information to the readers. However, the very old population is not suitable for evaluation of correlation of sarcopenia and mortality.
- Authors mentioned in several parts that “if the patients already has confirmation of sarcopenia, it is too late. (line 34-35)”, “Early diagnosis and treatment is advised when sarcopenia is suspected in order to try to reduce mortality (line 286-287)” and so on. However, the statements are too strong because the patients they enrolled in the present study are already very elderly. If authors would like to mention the early diagnosis and examine if the early diagnosis of sarcopenia improve the mortality, they should enroll lower age population and conduct the long term observation more than 2 years.
- Please change the title. The present title seems too broad and obscure to point out what authors show in this manuscript.
- Authors showed the clinical parameters and compared them between surviving patients and dead patients in the present study. CRP level seems higher in dead patients than that of surviving patients. Previous studies have shown that pro-inflammatory cytokines are linked to the mortality in dialysis population and measuring CRP seems easy to perform in clinical setting. Authors should explain the advantages of sarcopenia-related variables than CRP levels to predict the mortality.
- In discussion, authors mentioned the assessment of muscle mass by BIA is not accurate due to overhydration in hemodialysis patients. However, in different paragraph, ASM assessed by BIA is a good marker for the mortality in hemodialysis patients. Looks like there is a discrepancy between those statements. Please explain and keep consistency in authors’ statements.
- There are many typos to revise such as ASM; appendicular skeletal muscle mass, and AMS in line 161.
- In Figure2, authors should clarify the abbreviation and put more explanation. Does ASM criteria means ASM + GSD like Figure1B? Does GS criteria indicate ASM + GSD + GS? Please clarify.
Author Response
This paper entitled ” Sarcopenia according to EWGSOP2 and mortality in older hemodialysis patients” demonstrated that evaluation and definition of sarcopenia according to EWGSOP2 can predict the mortality in elderly patients with hemodialysis. This paper looks interesting, but there seems several questions remain to be resolved.
Major concern,
- Authors should highlight novelty of the present manuscript.
R: The novelty of this study is that it analyses the use of a new diagnostic algorithm for sarcopenia recently published (EWGSOP2) in a specific population group that has been little studied. There is limited information on the prevalence of sarcopenia in the very elderly on haemodialysis [(Lopes MB, et al] and the feasibility of using the EWGSOP2 definition and cut-off points in routine clinical practice. Progressive loss of muscle mass has been related to mortality in haemodialysis patients and loss of muscle mass has also been studied[(Gracia-Iguacel C,et al.], In our previous studies, the concept of muscle mass loss was not associated with any specific baseline value. The EWGSOP2 proposes cut-off points for muscle mass to define the confirmation of sarcopenia. In our study, these criteria represent well the prognosis in the studied group. The confirmation (ASM) and severity ( GS ) concepts proposed by EWGSOP2 are predictive values for mortality in the studied population.
This idea is now expanded upon in the discussion
- Authors enrolled sixty hemodialysis patients in the present study. Average age of the patients is 81.85 years old which is higher than the men’s life expectancy (79.59 yo) in Spain. Assessment of sarcopenia in the patients seems to provide interesting information to the readers. However, the very old population is not suitable for evaluation of correlation of sarcopenia and mortality.
R: We acknowledge the reviewer's comment and he is right that it would be interesting to analyse other age groups within haemodialysis patients. This initial study was planned for the older population group as we were observing a discrepancy between the presence of muscle mass determined by BIA and the deteriorated functionality of our patients, and we wanted to see how this was related to mortality in this population group.
The lack of analysis in other population groups has now been included as a limitation.
- Authors mentioned in several parts that “if the patients already has confirmation of sarcopenia, it is too late. (line 34-35)”, “Early diagnosis and treatment is advised when sarcopenia is suspected in order to try to reduce mortality (line 286-287)” and so on. However, the statements are too strong because the patients they enrolled in the present study are already very elderly. If authors would like to mention the early diagnosis and examine if the early diagnosis of sarcopenia improve the mortality, they should enroll lower age population and conduct the long term observation more than 2 years.
R: Indeed, the reviewer is right and it would have been interesting to analyse other age groups.
The age of dialysis patients in Spain has increased. Many of the patients enter at an advanced age with a good clinical situation, but our group has noticed a rapid deterioration, and is studying how to avoid it. We think that knowing the existence of sarcopenia can help to know the prognosis, and above all it could be avoided. In investigating this problem, we have encountered this particular situation.Elderly population studied have discrepancy in the use of the new diagnostic algorithm was most evident and had functional limitation but the muscle mass determined by BIA was adequate. Thus, according to our results, 60% of the patients retained muscle mass according to the EWGSOP2 diagnostic criteria, although they had lost 75-88% of their strength, depending on the criteria, and had lost 37-75% of their function, depending on the test chosen.
- Please change the title. The present title seems too broad and obscure to point out what authors show in this manuscript.
R: The title has been replaced by the running title to avoid to include acronyms in the title Sarcopenia and mortality in older hemodialysis patients
- Authors showed the clinical parameters and compared them between surviving patients and dead patients in the present study. CRP level seems higher in dead patients than that of surviving patients. Previous studies have shown that pro-inflammatory cytokines are linked to the mortality in dialysis population and measuring CRP seems easy to perform in clinical setting. Authors should explain the advantages of sarcopenia-related variables than CRP levels to predict the mortality.
R: This is clearly an established fact. In our study the CRP has been measured in our population but it is a worse predictor of mortality than the studied variables, therefore it was not included. This the high CRP hazard ration was 1.0 (0.8-1.5, p=0.045) while the risk in patients by muscle mass (ASM) was 2.5 (1.13-5.55, p=0.024) and the risk in those who had lost functionality was 3.02 (1.09-8.32, p=0.033).
We believe that inflammation is one of the underlying causes of poor muscle function.
- In discussion, authors mentioned the assessment of muscle mass by BIA is not accurate due to overhydration in hemodialysis patients. However, in different paragraph, ASM assessed by BIA is a good marker for the mortality in hemodialysis patients. Looks like there is a discrepancy between those statements. Please explain and keep consistency in authors’ statements.
R: It has now been corrected to convey the idea more clearly.
BIA is an easily accessible and inexpensive method for measuring muscle mass in patients on renal replacement therapy[33] and our group has already shown that the decrease in muscle mass assessed by BIA is a good marker of mortality in patients with chronic kidney disease[34,35]. It is a good marker only when muscle mass decreases, it is therefore a dynamic concept of evolution, or loss. Our current study shows that the exclusive use of BIA to diagnose sarcopenia does not assess muscle strength. Overhydration may alter the estimation of muscle mass by BIA in haemodialysis patients and therefore reduce the number of patients classified as sarcopenic[8,9]. However, our results shown also, that sarcopenia diagnosis with ASM by BIA with the values set by EWGSOP2 in isolation, is a good marker of mortality, likely not only due to objective loss of muscle mass, but also due to overhydration[36] or body composition [37]. ASM is possibly a poor marker of sarcopenia in elderly dialysis patients, but EWSGOP2 cut-off values seem to be good predictors of mortality.
- There are many typos to revise such as ASM; appendicular skeletal muscle mass, and AMS in line 161.
R: We have corrected several typographical errors in the text.
- In Figure2, authors should clarify the abbreviation and put more explanation. Does ASM criteria means ASM + GSD like Figure1B? Does GS criteria indicate ASM + GSD + GS? Please clarify.
R: Yes, the + sign involved both conditions (&)
Reviewer 2 Report
- Abstract
Please provide an explanation of the abbreviation.
Handgrip strength by dynamometry; GSD
Appendicular skeletal muscle mass; ASM
Gait speed; GS
- Materials and Methods; Primary variables studied
When was muscle strength (hand grip strength, test sit to stand to sit 5) and physical capacity (gait speed, timed-up and go test, short physical performance batter) measured?
After dialysis?
Before dialysis?
- Materials and Methods; Primary variables studied
How about the reproducibility of the measurement of MALTRON® brand BioScantouch i8 BIA? Can BioScantouch measure skeletal muscle mass with high accuracy?
- Materials and Methods; Statistics
Please indicate the adjust factors that were input in the Cox proportional hazard model.
For example, age, diabetes, etc.
- Result; Table 2, Table 4
Please add information on cachexia, peripheral arterial disease, malnutrition, physical activity.
Associated with death and sarcopenia, or gait speed.
- Results; Table 5
The information of hand grip strength, appendicular skeletal muscle mass, and gait speed is shown in Table 5. However, the information of test sit to stand to sit 5 and Timed-Up and Go test, Short Physical Performance Batter is not shown in Table 5.
Why is the information of test sit to stand to sit 5, and Timed-Up and Go test, Short Physical Performance Batter not shown?
- Results; Figure 2
Please analyze the following criteria in the Cox proportional hazards model.
Input variable
STS-5
GSD and/or STS-5
STS-5+ASM
GSD and/or STS-5+ASM
STS-5+ASM+GS
GSD and/or STS-5+ASM+GS
GSD+ASM+TUG
STS-5+ASM+TUG
GSD and/or STS5 +ASM+TUG
GSD+ASM+SPPB
STS-5+ASM+SPPB
GSD and/or STS-5+ASM+SPPB
GSD and/or STS-5+ASM+
GS and/or TUG and/or SPPB
- Discussion
Patients with loss of grip strength, probable sarcopenia, do not have higher mortality, nor do they have a higher risk of dying. However, loss of muscle mass confirmed sarcopenia and severe sarcopenia increased mortality risk.
The discussion of differences in mortality risk and muscle mass, grip strength, and gait speed is unclear.
- Limitation
The sample size is small.
Author Response
- Abstract
Please provide an explanation of the abbreviation.
Handgrip strength by dynamometry; GSD
Appendicular skeletal muscle mass; ASM
Gaitspeed; GS
R: The 3 parameters are referenced in the text and the description of GSD and GS was removed for the sake of brevity. A brief explanation is included again. ASM is referred to in detail in the conformation section.
- Materials and Methods; Primary variables studied
When was muscle strength (hand grip strength, test sit to stand to sit 5) and physical capacity (gait speed, timed-up and go test, short physical performance batter) measured?
Afterdialysis?
Beforedialysis?
R: We thank the reviewer for this comment. We now add in the method when the strength measurements and physical tests were performed.
The strength measurements and physical tests were performed on the second day of the week before the dialysis treatment. In this way we tried to avoid being influenced by the fatigue experienced by dialysis patients after treatment.
- Materials and Methods; Primary variables studied
How about the reproducibility of the measurement of MALTRON® brand BioScantouch i8 BIA? Can BioScantouch measure skeletal muscle mass with high accuracy?
R: The reproducibility of the BIA has been proven in previous studies [Sarcopenia assessed by 4-step EWGSOP2 in elderly hemodialysis patients: Feasibility and limitations]. ASM is an estimate made by tetrapolar BIAs, but over time it has been consistent with the other studies conducted.
- Materials and Methods; Statistics
Please indicate the adjust factors that were input in the Cox proportional hazard model.
For example, age, diabetes, etc.
R: Multivariate Cox regression analysis was performed with those parameters that were significant between death/survival. No adjustments were applied because the aim was to see how the different factors determining sarcopenia interacted with each other and secondly because the population was small for this type of analysis.
- Result; Table 2, Table 4
Please add information on cachexia, peripheral arterial disease, malnutrition, physical activity.
Associated with death and sarcopenia, or gait speed.
R: In this study we have not assessed cachexia, peripheral vascular involvement or physical activity. The main reasons for not assessing physical activity are: how can physical activity be assessed for comparison, if we are assessing tests that in themselves do this. Malnutrition has not been assessed in a complete way. There are collected values such as folds, cholesterol and albumin. We are trying to carry out a specific study linking malnutrition and sarcopenia, but this requires a separate analysis.
- Results; Table 5
The information of hand grip strength, appendicular skeletal muscle mass, and gait speed is shown in Table 5. However, the information of test sit to stand to sit 5 and Timed-Up and Go test, Short Physical Performance Batter is not shown in Table 5.
R: At the reviewer's suggestion we expanded table 5 and added its data to the text accordingly.
- Results; Figure 2
Please analyze the following criteria in the Cox proportional hazards model.
Input variable
STS-5
GSD and/or STS-5
STS-5+ASM
GSD and/or STS-5+ASM
STS-5+ASM+GS
GSD and/or STS-5+ASM+GS
GSD+ASM+TUG
STS-5+ASM+TUG
GSD and/or STS5 +ASM+TUG
GSD+ASM+SPPB
STS-5+ASM+SPPB
GSD and/or STS-5+ASM+SPPB
GSD and/or STS-5+ASM+
GS and/or TUG and/or SPPB
R: Although, as mentioned above, the different sarcopenia determinants interacted with each other, at the reviewer's request we include the various values and combinations presented in table 3.
- Discussion
Patients with loss of grip strength, probable sarcopenia, do not have higher mortality, nor do they have a higher risk of dying. However, loss of muscle mass confirmed sarcopenia and severe sarcopenia increased mortality risk.
The discussion of differences in mortality risk and muscle mass, grip strength, and gait speed is unclear.
R: Although loss of grip strength has been related to increased mortality in different studies, in our study, with the EWGSOP2 criteria and in this population, it is not a prognostic criterion. But the EWGSOP2 criteria for muscle mass and gait speed are clear prognostic values for mortality. When muscle mass decreases, and especially functionality, the risk of death increases.
- Limitation
The sample size is small.
R: We agree with the reviewer and this comment is now added in limitations
Round 2
Reviewer 1 Report
I have no further concern.
Author Response
I have no further concern.
R: Thank the reviewer for his comments and we would like to mention the various modifications we have made to the manuscript in this round of revision to make the message clearer.
Reviewer 2 Report
- Materials and Methods; Primary variables studied
Revision 2 comment: Please add the following sentence to the text (Materials and Methods).
Author comments: The reproducibility of the BIA has been proven in previous studies [Sarcopenia assessed by 4-step EWGSOP2 in elderly hemodialysis patients: Feasibility and limitations]. ASM is an estimate made by tetrapolar BIAs, but over time it has been consistent with the other studies conducted.
- Materials and Methods; Statistics
Please indicate the adjust factors that were input in the Cox proportional hazard model.
Author comments: Multivariate Cox regression analysis was performed with those parameters that were significant between death/survival. No adjustments were applied because the aim was to see how the different factors determining sarcopenia interacted with each other and secondly because the population was small for this type of analysis.
Revision 2 comment: Age and cardiovascular disease are significantly associated with mortality risk. Therefore, multivariate Cox regression analysis should be performed with age and cardiovascular disease added as adjustment factors.
In addition, multivariate Cox regression analysis should be performed with a larger sample size.
- Result; Table 2, Table 4
Please add information on cachexia, peripheral arterial disease, malnutrition, physical activity.
Revision 2 comment: Please add the following sentence to the text (Limitation).
Author comment: In this study we have not assessed cachexia, peripheral vascular involvement or physical activity.
Author Response
- Materials and Methods; Primary variables studied
Revision 2 comment: Please add the following sentence to the text (Materials and Methods).
Author comments: The reproducibility of the BIA has been proven in previous studies [Sarcopenia assessed by 4-step EWGSOP2 in elderly hemodialysis patients: Feasibility and limitations]. ASM is an estimate made by tetrapolar BIAs, but over time it has been consistent with the other studies conducted.
R: The abovementioned phrase has been included in the discussion
- Materials and Methods; Statistics
Please indicate the adjust factors that were input in the Cox proportional hazard model.
Author comments: Multivariate Cox regression analysis was performed with those parameters that were significant between death/survival. No adjustments were applied because the aim was to see how the different factors determining sarcopenia interacted with each other and secondly because the population was small for this type of analysis.
Revision 2 comment: Age and cardiovascular disease are significantly associated with mortality risk. Therefore, multivariate Cox regression analysis should be performed with age and cardiovascular disease added as adjustment factors.
In addition, multivariate Cox regression analysis should be performed with a larger sample size.
R: The figure 2 has been replaced by the analysis with the suggested variables and the text has been changes accordingly. The original figure has been moved to supplementary. According to the referee suggestions, changes have been added to the results and discussion.
- Result; Table 2, Table 4
Please add information on cachexia, peripheral arterial disease, malnutrition, physical activity.
Revision 2 comment: Please add the following sentence to the text (Limitation).
Author comment: In this study we have not assessed cachexia, peripheral vascular involvement or physical activity.
R: The limitation has been included